# Immunotherapies for Alzheimer’s Disease—A Review

**DOI:** 10.3390/vaccines10091527

**Published:** 2022-09-14

**Authors:** Zachary Valiukas, Ramya Ephraim, Kathy Tangalakis, Majid Davidson, Vasso Apostolopoulos, Jack Feehan

**Affiliations:** 1College of Health and Biomedicine, Victoria University, Melbourne, VIC 3011, Australia; 2Institute for Health and Sport, Victoria University, Melbourne, VIC 3021, Australia; 3First Year College, Victoria University, Melbourne, VIC 3011, Australia; 4Institute for Sustainable Industries and Liveable Cities, Victoria University, Melbourne, VIC 3011, Australia; 5Immunology Program, Australian Institute for Musculoskeletal Science (AIMSS), Melbourne, VIC 3021, Australia

**Keywords:** Alzheimer’s disease, immunotherapies, vaccines, amyloid-beta, dementia

## Abstract

Alzheimer’s disease (AD) is a chronic neurodegenerative disorder that falls under the umbrella of dementia and is characterised by the presence of highly neurotoxic amyloid-beta (Aβ) plaques and neurofibrillary tangles (NFTs) of tau protein within the brain. Historically, treatments for AD have consisted of medications that can slow the progression of symptoms but not halt or reverse them. The shortcomings of conventional drugs have led to a growing need for novel, effective approaches to the treatment of AD. In recent years, immunotherapies have been at the forefront of these efforts. Briefly, immunotherapies utilise the immune system of the patient to treat a condition, with common immunotherapies for AD consisting of the use of monoclonal antibodies or vaccines. Most of these treatments target the production and deposition of Aβ due to its neurotoxicity, but treatments specifically targeting tau protein are being researched as well. These treatments have had great variance in their efficacy and safety, leading to a constant need for the research and development of new safe and effective treatments.

## 1. Introduction

Alzheimer’s disease (AD) is a chronic neurodegenerative disorder and the most prevalent form of dementia, accounting for 60–80% of cases [1] and is a growing burden on healthcare systems nationally and internationally. The Global Burden of Disease Study estimated that 57.4 million people globally were living with dementia in 2019 and this is predicted to reach 152.8 million people by 2050 [2]. Dementia is an umbrella syndrome characterised primarily by progressive deterioration in multiple cognitive domains, causing impairment in daily functioning across social, physical, and professional spheres [3,4]. The elderly are those primarily affected by AD [5], though this is not necessarily a direct outcome of ageing [6]. The neuropathology of AD is underpinned by the accumulation of plaques, which are extracellular aggregates of amyloid-β (Aβ), and neurofibrillary tangles (NFTs), which are intracellular aggregates of tau protein.

Currently, people diagnosed with AD are treated with acetylcholinesterase inhibitors (rivastigmine, galantamine, donepezil) and an N-methyl-D-aspartate (NMDA) receptor antagonist (memantine). Though these treatments are able to restrain the manifestations of dementia for a period, they are incapable of completely halting either disease progression or symptom manifestation [7], and are hampered further by a number of common side effects, such as gastrointestinal irritation, dizziness, and headache [8]. Since AD is a convoluted disease caused by multiple components, its aetiology and pathogenesis remain obscure, and existing single-target, single-action drugs cannot radically delay its progression [9].

More recently however, Aβ immunotherapy has gained attention as a promising approach to modify the course of AD. Immunotherapies use synthetic peptides or monoclonal antibodies (mAbs) to decrease the Aβ load in the brain and slow the progression of the disease, by inducing the immune system to break down and clear the aberrant proteins [10]. In addition to these passive immunotherapies, the development of vaccines against target proteins in AD may have beneficial outcomes against the disease. Herein, we discuss the pathophysiology of AD and how it can be used to target effective immunotherapies in an attempt to prevent and manage AD. We also evaluate the current progress in immunotherapies/vaccines against AD, providing recommendations for future work.

## 2. Pathophysiology of Alzheimer’s Disease

Neuropathological examination of individuals living with AD identifies dense protein aggregates comprising extracellular Aβ plaques and intracellular NFTs (Figure 1) [11]. Examination of these individuals also presents associated chronic inflammation in the affected areas of the brain [8]. The tau found in the intracellular NFTs is a microtubule stabilising protein that plays an important role in Aβ toxicity, with levels of the protein in the brain correlating strongly with the cognitive decline seen in AD patients [12,13]. This is caused by an irreversible phenomenon of neurodegeneration and apoptosis in the hippocampal and entorhinal cortex regions, leading to difficulty with memory, loss of executive functioning, apathy, and depression [8,14]. Adverse mechanisms such as impairment of brain metabolic function [15], blood–brain barrier (BBB) disruption [16], increased oxidative stress, calcium homeostatic disturbance, impairment of cellular autophagy, neuroinflammation, and neuronal apoptosis commonly co-exist, contributing to the aetiology of the disease [8,17]. Due to the neurovascular coupling, the normal bulk clearance of cerebrospinal fluid (CSF) and interstitial fluid (ISF) becomes hindered [18].

### 2.1. BACE-1 and Aβ Generation

The amyloid hypothesis is the most extensively studied concept of AD, in which abnormal processing of Aβ and/or impairment of its systemic clearance may be responsible for the progression of AD-related phenotypes (Figure 2) [19]. The characteristic Aβ plaques seen in AD originate from proteolysis of the amyloid precursor protein (APP) by sequential enzymatic action of beta-site amyloid APP-cleaving enzyme 1 (BACE-1) [20]. The membrane-bound APP is first cleaved by BACE-1, generating soluble amyloid precursor protein β (sAPPβ) and an integral fragment called C99 [21]. C99 is then cleaved again by γ-secretase [6], generating amyloid precursor protein intracellular domain (AICD) and free Aβ, an insoluble 36–43 amino acid peptide, which aggregate to form oligomers.

These oligomers are thought to acts as ‘seeds’ which induce further Aβ mis-folding and aggregation [22], and importantly are neurotoxic, leading to the apoptosis of neurons locally, and consequently the characteristic symptomatology of AD. The exact functions of APP and APP-derived fragments are not fully understood [23], and BACE-1 knockout animals appear physiologically healthy and do not express Aβ [24]. BACE-1 has also been shown to have increased concentrations and rates of activity in AD brains and body fluids, supporting the hypothesis that BACE-1 may play a significant role in AD [25]. Therefore, BACE-1 is a prime drug target for slowing down Aβ production in early AD [26,27], however, clinical trials have thus far had disappointing results, with a number of phase II and III studies halted early for limited or no effect on symptoms and variable effects on Aβ volume or deposition, with association with a reduction in brain tissue volume in follow-up examinations [28,29,30].

### 2.2. Tau Protein

One of the hallmarks of AD is the presence of NFTs, and filamentous inclusions in pyramidal neurons [31]. Tau proteins are responsible for several key functions within the central nervous system (CNS), primarily being stability modulators of axonal microtubules. Like Aβ oligomers, intermediate aggregates of abnormal tau molecules are cytotoxic and impair cognition [6]. While these NFTs have long been associated with AD, whether they are a pathogenic driver of the disease, or a result of Aβ or other underlying mechanisms, is unclear. In early-stage AD mouse models, it was noted that site-specific phosphorylation of tau inhibited Aβ toxicity via the neuronal p38 mitogen-activated protein kinase (MAPK) p38γ, a key signalling molecule involved in cellular stress responses [32].

### 2.3. APOE-ε4

The apolipoprotein E-epsilon 4 (APOE-ε4) genotype has been implicated as a risk factor for late-onset AD [33], with a synergistic role of APOE-ε4 and inflammation, alongside vascular factors, as a possible pathway to the onset of AD [34]. The human APOE gene has three key isoforms: APOE2, APOE3, and APOE4, with corresponding proteins [35,36]. Carrying and expressing the APOE4-coding allele is the chief genetic risk factor for AD, with predictive values exceeding polygenic scores for cognitive ageing in elderly populations [35,37]. APOE4 status has also been linked to Parkinson disease-associated dementia [38]. It has been suggested that the effect of APOE4 in the meningeal lymphatic system could reveal a missing link in our understanding of the aetiology and pathology of AD [33,35].

### 2.4. TREM2

Triggering receptor expressed on myeloid cells 2 (TREM2)—a marker of microglial inflammatory reactions—is another important marker in the pathophysiology of AD [39]. Soluble TREM2 (sTREM2) is released upon microglial activation, leading to increased levels of CSF sTREM2 seen in AD [40], and is involved in APOE4′s downstream activation of microglia [35]. TREM2 also facilitates additional microglial activation and clustering around Aβ and NFTs, increasing amyloid uptake, phagocytic activity, and plaque compaction in the early stages of AD [41].

### 2.5. Other Contributing Factors

Other factors such as smoking, reduced physical activity, infection, and prior conditions (e.g., diabetes and obesity) can also lead to developing AD [42,43], with likely mechanisms involving abnormal cholesterol metabolism and chronic inflammation. A humoral immune component has also been implicated in the pathology of AD [44]. It is now widely accepted that circulating immune cells have a significant role in brain pathologies and that their impact is dependent on their type, location, and activity [45].

## 3. Role of the Immune System in AD

The adaptive immune system is central to the pathogenesis and progression of AD, with glial and T lymphocyte interactions a key driver of neuroinflammation (Figure 3) and neuronal destruction. Microglia and astroglia, the brain-resident immune cells, are powerful regulators of neuroinflammatory responses in AD [46,47]. Microglia are the principal immune effector in the CNS, acting as both phagocytes and antigen-presenting cells (APCs) and there has been some debate regarding their contribution to the clearance of Aβ following their activation [48]. Recent advances in neuroinflammation research has led to the discovery of several novel inflammatory pathways regulating many cerebral pathologies, such as the 5-lipoxygenase (ALOX5) pathway [8].

The lymphatic system of the brain carries immune cells from the CSF, connecting to the deep cervical lymph nodes, which enables peripheral T cells to respond to brain antigens [49]. Both CD4^+^ T helper and CD8^+^ effector T lymphocytes aggregate in the brain in AD, and play a role in the pathology and progression of the disease [50]. However, in contrast to the peripheral mechanisms, the major APCs in AD are the microglial cells, which show increases in genes and markers associated with T cell interaction [51]. There also appears to be a loss of intrinsic immunosuppression associated with AD, with transient depletion of Foxp3^+^ regulatory T cells affecting the choroid plexus and associated with subsequent recruitment of immunoregulatory cells, such as monocyte-derived macrophages and regulatory T cells, to cerebral sites of plaque pathology [52]. These effector and regulatory functions of lymphocytes are altered with ageing, and other immune manifestations accompany the progression of AD [53]. The neurodegeneration and concurrent involvement of the peripheral immune system in AD patients has been suggested to promote leukocyte division and telomere shortening [54]. 

This makes the diagnosis of novel therapeutic interventions of critical importance in AD management moving into the future. For decades, therapies have been developed that directly target Aβ production or aggregation, however, all have failed to slow disease progression [12].

### Cytokines in AD

While microglia and astrocytes play numerous roles within brain tissue, their involvement in neuroinflammation through cytokine activity is a crucial aspect of AD that requires significant management. As microglia are recruited and overactivated and astrocytes detect cellular death, they release pro-inflammatory cytokines into the surrounding extracellular space which then have a range of actions. Tumor necrosis factor α (TNF-α) is highly prolific, being the most studied cytokine involved in AD. In transgenic mouse models of AD, elevated TNF-α levels were observed in brain tissues and correlated with levels of cognitive decline in the mice. Furthermore, deletion of the tumor necrosis factor receptor 1 (TNFR1) gene in transgenic AD mice showed decreased Aβ generation, plaque burden, BACE-1 expression, and cognitive deficits [55]. Evidence also suggests that TNF-α directly interferes with microglial clearance of Aβ deposits [56]. Other cytokines also play a role in neuroinflammation. Interleukin (IL) 1α plays a significant role in overexpression of APP, as well as being highly expressed in an AD brain compared to a healthy brain [57] which can create a loop of IL-1 secretion and APP synthesis. IL-6 is also an important molecule within AD. While its direct impact is yet to be properly understood, it is established to be involved in an upregulation of TNF-α and microglial activation [58]. Moreover, blocking IL-6 activity can improve long-term memory and hippocampal function [59]. Lastly, inhibition of the IL-12/IL-23 pathway may attenuate AD pathology and cognitive deficit. In a pathway that is not yet understood, ablation of the p40 subunit within IL-12/23 was shown to have reduced the amount of soluble Aβ and improved cognitive function in AD mouse models [48,60].

## 4. Immunotherapies for AD

Immunotherapies have become one of the most promising methods to reverse or slow the progression of AD [61]. Several types of Aβ peptide immunotherapy for AD are under investigation, through approaches such as active immunisation and mAbs directed against Aβ peptide [62] and tau pathology [63,64,65,66]. Focus has been on the development of multitarget AD immunotherapies, the optimisation of antibody titers and epitopes, pharmacogenetic/pharmacoepigenetic validation of the immunisation procedure, the prophylactic treatment of genetically stratified patients at a pre-symptomatic stage, and the definition of primary endpoints in prevention, based on objective/multifactorial biomarkers [64]. Matrix metalloproteinases’ involvement in CNS disorders, such as AD, has also made them attractive therapeutic targets [67].

### 4.1. Antibody Therapies for AD

In clinical trials of patients with early AD, administration of anti-amyloid antibodies reduced plaque volume, suggesting that passive immunotherapies may be promising disease-modifying interventions (Figure 4). Currently, the only approved disease-modifying treatment of AD is the drug aducanumab, a mAb specific to Aβ and that shows efficacy in the reduction in Aβ density within patients [68]. Single chain fragment variables (scFvs), containing only the variable region of the heavy and light chains of antibodies, have shown great potential for the treatment of AD [69]. Thirteen phase III trials using the mAbs bapineuzumab [70], solanezumab [71], gantenerumab [72,73], and crenezumab [74] have been conducted in recent years, however, all were discontinued due to a lack of efficacy on improving cognitive function (Table 1). Another candidate, BAN2401, also known as lecanemab, entered a phase III clinical trial in July 2020 and displays significant reduction in Aβ aggregates and improvement in clinical symptoms [75,76]. A post-translationally modified variant of the Aβ peptide which has a pyroglutamate at the N-terminus (pGlu3) is an attractive antibody target, due to its neo-epitope character and its propensity to form neurotoxic oligomeric aggregates [77]. PBD-C06 is an antibody targeting pGlu-Aβ which also circumvents inflammatory issues (complement inactivation) and immunogenicity (de-immunisation) and has great potential to clear the most toxic Aβ aggregates and improve cognition in AD patients at effective doses, while also avoiding inhibition of inflammatory responses in vitro [77].

### 4.2. Active Vaccinations against AD

There are a number of targets currently being evaluated for use in an active vaccine therapy for AD (Figure 4). About 140 (85%) immunisation procedures against Aβ deposition and 25 (15%) against tau have been reported, but no Food and Drug Administration approval of any AD vaccine has been achieved [64]. An Aβ_42_ trimer DNA vaccine may provide a path forward in finding viable options for AD prevention or a means of delaying disease progression. The DNA vaccine, AV-1959D, targeting the N-terminal epitope of the Aβ peptide, is immunogenic in mice, rabbits, and non-human primates, as well as being effective in mouse models of AD (Table 2). Repeated dose safety assessment did not find any adverse short- or long-term effects of the vaccine in mice. Mice treated with the vaccine demonstrated elevated anti-Aβ antibodies over time [79]. Early immunisation with a conjugated Aβ_3–10_-keyhole limpet hemocyanin vaccine can greatly reduce tau phosphorylation, however, these immunotherapies are not clinically effective when administered too late [61].

Vaccination targeting only the tau protein has shown benefits in some mouse studies but human studies are limited [65]. To prevent the accumulation of plaques, novel and safer plant-based vaccine strategies have been suggested [84]. In 2002, the first active AD vaccine (AN1792) developed by ELAN in Ireland and Wyeth in the USA went through a phase IIa clinical trial but was suspended due to the development of meningoencephalitis in ~6% of the individuals [10,80]. The exact mechanism of this was unknown and determined to have no clear relation to serum anti-Aβ_42_ antibody titers but may have been an autoimmune response rising from T cell interactions [85,86]. Other groups have evaluated the effect of combining systemic immunomodulators and influenza vaccines as a means of increasing immune action against plaques. As such, programmed cell death protein 1 (PD-1) checkpoint blockade—inhibition of T cell apoptosis by preventing binding to PD-1, known to modify AD [87]—in conjunction with the influenza vaccine, is hypothesised to have a dual immunostimulatory effect that could provide clinical benefit. The combination treatment was effective in attenuating cognitive deficit and Aβ pathology build-up in APP/PS1 mice through recruitment of monocyte-derived macrophages to the CNS [88]. More recently, a vaccine developed by modifying yeast cells to express Aβ1-15 on their cell wall, named Y-5A15, was shown to improve cognitive function, and decrease plaque formation and neuronal damage in animal models [78]. 

In November of 2021, a phase I clinical trial of the vaccine Protollin was reported [89]. Protollin is a combination of *Neisseria meningitidis* outer membrane proteins complexed with *Shigella flexneri* 2a lipopolysaccharide. This combination works by activating Toll-like receptors (TLRs) 2 and 4 within the nasal cavity. The immune response then moves to the cervical lymph nodes where CD4^+^ T cells can be activated and migrate to the CNS [81]. The vaccine is delivered intranasally and displays efficacy in the removal of Aβ in transgenic mouse models of AD [87,88].

### 4.3. The Limitations and Challenges of Immunotherapies

While immunotherapies demonstrate a promising route towards the treatment of AD, there are problems and complications that can arise. As described with the vaccine AN1792, the development of adverse events or reactions to the treatment is a major concern. Over-reactivity is a possibility when administering an immunotherapy and can result in more harm than good for the patient. In the case of AD, an over-reactive immune response could lead to further neurodegeneration via several pathways. Similarly to microglial and astrocyte signalling, excess inflammatory cytokine production could lead to further neuroinflammation that could exacerbate the disease state [90]. Furthermore, autoreactive T cell responses have the potential to develop and are a crucial safety consideration that must be monitored for.

Other concerns relating to efficacy of immunotherapies must also be considered. Part of the appeal of immunotherapies is that they rely on the patient’s immune system as the treatment for the condition. However, patients in advanced stage dementia are typically older individuals and present with weaker immune systems, either age-related or being immunocompromised due to comorbidities. As a result, there are several avenues by which an immunotherapy may fail. For example, decreased activity and availability of naïve CD4^+^ and CD8^+^ T cells [91], the increased likelihood of CD5^+^ B cells producing auto-antibodies [92], as well as the impaired production of naïve B cells within bone marrow [93]. Furthermore, age-related immunosenescence may contribute to a tolerance of the immunotherapy and it may be unable to elicit immunogenicity [94,95].

### 4.4. Analysis of Immunotherapeutic Efficacy

When conducting an immunotherapeutic trial, analysis of treatment efficacy is critical for the further development of that treatment. As a result, the use of concurrent biomarker analysis is recommended. Common biomarker analysis of AD looks for changes of CSF Aβ_42/40_, phosphorylated tau (P-tau) and total tau (T-tau) [96] and neurogranin, a cerebral post-synaptic protein involved in long-term potentiation, whose elevation in the CSF appears specific to AD [97].

### 4.5. Cerebral Amyloid Angiopathy and Amyloid-Related Imaging Abnormalities

Cerebral amyloid angiopathy (CAA) and amyloid-related imaging abnormalities (ARIAs) are conditions that present great challenges for immunotherapies, particularly with mAb treatments. CAA is the deposition of Aβ within cerebral vascular tissue and is heavily implicated in intracerebral haemorrhages and ARIA complications and is common among AD patients [98]. In clinical trials, aducanumab and lecanemab exhibited strong ARIA complications, which may have been related to CAA. The treatments resulted in ARIA–vasogenic oedema (ARIA-E), which was more common in participants that were APOE-ε4 positive [68,76], further demonstrating the gene’s implication in AD [37]. Furthermore, a common trend within mAb trials is the demonstration of dose-dependency for the removal of Aβ. Both aducanumab and lecanemab demonstrated the greatest reduction in Aβ within the 10 mg/kg dose groups. Unfortunately, the presentation of ARIA complications has a negative impact on the appeal of the treatment. Another mAb—crenezumab—demonstrated fewer ARIA complications. In the trial, one participant receiving a 15 mg/kg dose every four weeks (n = 247) exhibited asymptomatic ARIA-E, and one participant in the same cohort exhibited asymptomatic ARIA–cerebral macrohaemorrhage (ARIA-H) [74]. While the treatment was tolerated better, presented lower safety implications, and reduced Aβ density, the primary focus of the study—cognition—was not met.

## 5. Conclusions

Although Aβ is the most extensively studied pathological hallmark of AD pathophysiology, many recent therapeutic approaches directing against this peptide have often failed in clinical trials, and thus, more attention is shifting toward tau pathology and neuroinflammation as therapeutic targets. Immunotherapy focusing on reducing the Aβ burden is a promising treatment strategy for AD [78]. This might be attributed to deficient pathogenic targets, inappropriate models, defective immunotherapeutic procedures, and inadequate clinical trial design. Two important factors that may have been under-estimated in AD pre-clinical research are the relevance of current AD mouse models and the immunological differences between mice and humans. The exact contribution of the different reactive microglia subtypes to AD is currently unclear and the subject of intense research. Many factors need to be considered—including sex, age, species, molecular diversity, health status, communication with the periphery—to fully decipher the role of microglia in AD. This is undoubtedly challenging but also a very exciting field of research, which holds the promise of defining innovative therapeutic strategies and subsequently reducing the socio-economic burden of this devastating disease [99]. Effective vaccines which halt or slow AD might be an effective and convenient approach to avoid enormous treatment-related expense [62]. A key consideration in vaccination against AD is the timing of treatment. Given that there is an age-related decline in immune function, vaccines may be more likely to prevent AD instead of providing treatment. Early vaccination, which prevents plaque build-up before symptoms have shown, may be more effective, while also providing a rationale for the current failure of AB immunotherapies in trials, as these are always tested in patients with symptomatic disease. 

A vaccine against AD is technically feasible; however, important methodological aspects should be changed for clinical success, including (i) the development of multitarget AD immunotherapies; (ii) the optimisation of antibody titers and epitopes; (iii) the pharmacogenetic/pharmacoepigenetic validation of the immunisation procedure; (iv) the prophylactic treatment of genetically stratified patients at a pre-symptomatic stage; and (v) the definition of primary endpoints in prevention, based on objective/multifactorial biomarkers. Even with exquisite protocols, an individual, uni-target vaccine would be potentially useful in at most 20–30% of defined cases, according to the genetic, epigenetic, and pharmacogenetic background of AD patients [64]. Compared to passive immunotherapies, vaccines have several disadvantages. They depend on some degree of consistency of immune response in each individual, but people are heterogeneous. The characteristics of antibodies induced by vaccines are limited by the human immune system and cannot, for example, include artificial modifications for which therapeutic mAbs might be given to optimise their effectiveness. However, passive therapies are costly, and short term, while vaccines produce antibodies internally at much lower cost, so vaccination might be the most promising approach to reducing the global burden of dementia [100].

## Figures and Tables

**Figure 1 vaccines-10-01527-f001:**
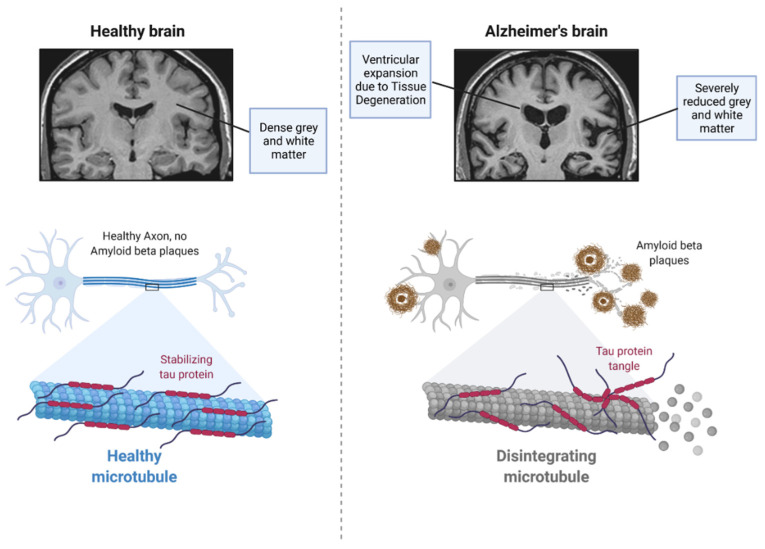
Pathophysiological differences between a healthy and an AD brain. Tissue degeneration is highly prevalent. Neuronal degeneration to Aβ and NFTs is highlighted. Created with BioRender.com.

**Figure 2 vaccines-10-01527-f002:**
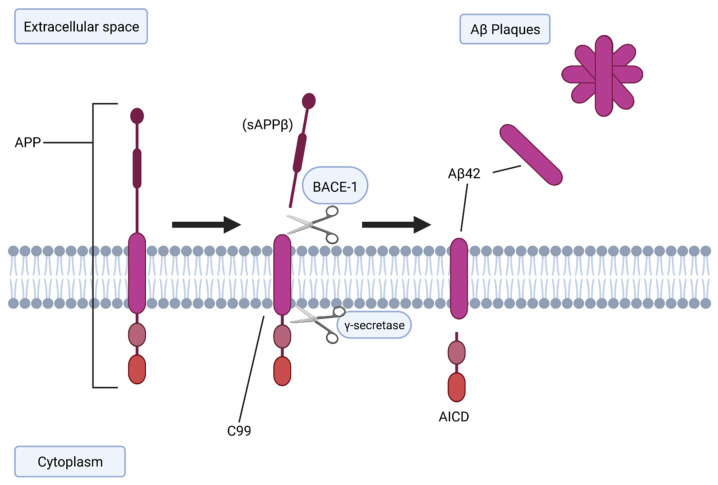
Generation of Aβ from APP. APP is cleaved by BACE-1 to generate C99 and is then cleaved again by γ-secretase to generate free A-beta. Created with BioRender.com.

**Figure 3 vaccines-10-01527-f003:**
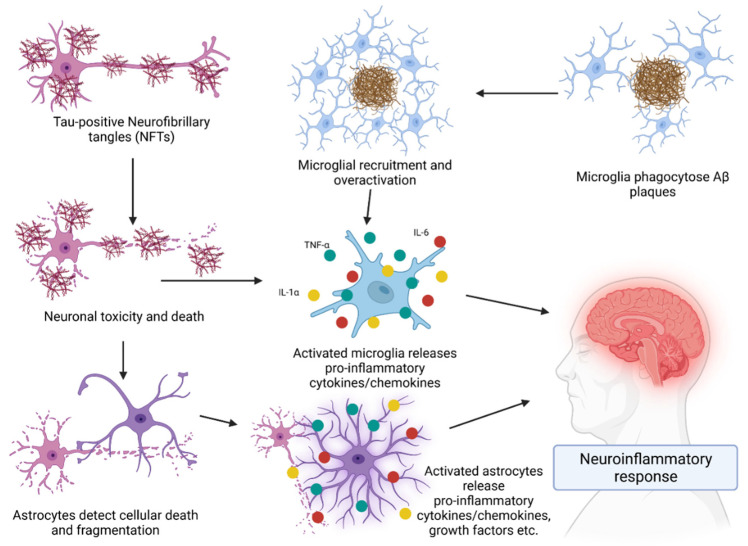
Glial- and astrocyte-mediated neuroinflammation related to AD. Both microglial and astrocyte-mediated pathways result in the release and activation of pro-inflammatory molecules such as IL-1α, IL-6, and TNF-α. Created with BioRender.com.

**Figure 4 vaccines-10-01527-f004:**
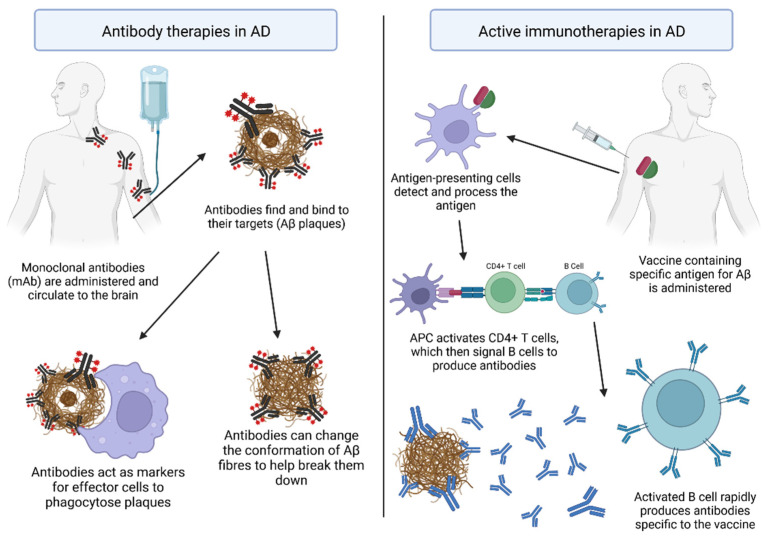
The general theory for immunotherapy against AD pertaining to Aβ pathophysiology. Antibody therapy involves the administering of antibodies to the patient, active immunotherapy involves using a vaccine to make the patient produce their own. Created with BioRender.com.

**Table 1 vaccines-10-01527-t001:** Clinical trial information for mAb treatments against AD and their findings. Treatments may have influenced Aβ deposition, but none reach the primary endpoint of ameliorating cognitive decline.

Ref.	Compound	Phase	Target	Type	Participants	Findings
[68]	Aducanumab	Ib	Aβ	mAb	197	Reduced Aβ, did not improve cognition.
[70,78]	Bapineuzumab	III	Aβ	mAb	1121, 1331	Did not improve cognition, did not reduce Aβ deposition.
[71]	Solanezumab	III	Aβ	mAb	2052	Did not improve cognition, Levels of Aβ_40_ decreased, Aβ_42_ did not change.
[73]	Gantenerumab	III	Aβ	mAb	799	Study halted due to no effect on cognition or Aβ deposition.
[74]	Crenezumab	II	Aβ	mAb	448	No effect on cognition, elevated CSF levels of Aβ were associated with treatment.
[76]	Lecanemab	II	Aβ	mAb	854	Treatment showed a reduction in Aβ and a reduction in cognitive decline over an 18-month period, missing 12-month primary endpoints.

**Table 2 vaccines-10-01527-t002:** Research information for several active vaccine treatments against AD and their findings. All demonstrate potential in the form of immune responses being generated but must be monitored carefully to prevent adverse events from occurring.

Ref.	Compound	Phase	Target	Type	Participants	Findings
[79]	AV-1959D	Pre-clinical	Aβ	DNA Vaccine	60	No short- or long-term toxicities demonstrated. The vaccine elicited an immune response in the form of antibody production specific to Aβ_42_
[78]	Y-5a15	Pre-clinical	Aβ	Vaccine	N/A	Treatment elicited significant levels of Aβ antibodies, reduced levels of Aβ, and improved cognitive function in mice.
[10,80]	AN1792	IIa	Aβ	Vaccine	375	Reduced Aβ load in the brain, terminated due to development of adverse events resulting from the treatment.
[79,81,82,83]	Protollin	Pre-clinical	Aβ	Vaccine	N/A	Significant reduction in Aβ in mice, cognitive function improved following treatment. Adjuvant was not observed in brain tissue.

## Data Availability

Not applicable.

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
