# Peer review of "Immunotherapies for Alzheimer’s Disease—A Review"

_vaccines, 2022, doi:10.3390/vaccines10091527_

Round 1

Reviewer 1 Report

The authors could provide possible etiology of AD in detail in this review article. Unfortunately, in the paragraph regarding to the role of the immune system in AD, the authors could not sufficiently describe about the cytokines which is important issue for etiology of AD, since some kinds of cytokines such as IL and TNFa which are released from over-activated microglia and astrocytes would trigger neuroinflammatory responses. I hope they would add the description about the cytokines in detail, which would improve this article more than present form.

Author Response

The authors appreciate your feedback and agree that cytokines should have had a more sufficient description given their role in AD and have added a section specific to them. Section 3.1 Cytokines in AD begins on line 169 and ends on line 190 and discusses TNF-α and several other inflammatory cytokines that are involved in AD and neuroinflammation.

Reviewer 2 Report

Understanding the immunopathogenesis of neurodegenerative and cerebrovascular diseases defines new targets and approaches to treatment. Suppression of neuroinflammation is advisable already at the early stages of cognitive decline, when the information content of routine clinical, laboratory and instrumental examination of patients is insufficient to clarify the causes of cognitive decline. The article summarizes modern ideas about the immunopathogenesis of AD. The mechanism of the development of neuroinflammation is presented as a cascade of successive events, eventually closing in a self-sustaining inflammatory response. Damage-associated molecular fragments and specific receptors, intracellular signal transduction in microglial cells, cytokines, and adhesion molecules are considered as potential points of application of immunomodulatory therapy and preventive vaccination. Information about the current level of development of immunotherapy for AD and further prospects for its use are given.

Author Response

We thank the reviewer for their positive feedback

Reviewer 3 Report

Dear Authors,

Indeed, it will be of good interests for other researchers to read the review on the immunotherapies in AD.

Since authors also included the BACE-1 inhibitors for comparison, the title should be changed to represent the BACE-1 inhibitors or drop the sections/tables on BACE-1 inhibitors, title is misleading the readers, which should be more specific towards A-beta.

There is on subtitle of the table, which should be separated into 2 tables, one for the mAb and the other one for vaccines.

To bring additional interests by the researchers, I recommend authors to include following information in the table, such as target epitope(s) of the mAb, mode of administrations, dosages, half-life, side effects, local or international trials and reasons for stopping the clinical trial.

All figures should be more specific for representing the APP processing, mode of administrations (mAbs does not get delivered as a pill) and mechanisms.

Authors need to discuss the difficulties and limitations of the immunotherapies extensively, instead of mentioning briefly.

Author Response

The authors appreciate your feedback and understand your comment regarding BACE-1. We agree and have changed the title of that section (Section 2.1 Lines 76-101) from “The Amyloid Hypothesis” to “BACE-1 and Aβ generation”. The authors felt that this was a more appropriate title as it now covers the prominent role of BACE-1 within AD while also within the context of Aβ. Related to this, the BACE-1 targeting drugs within the table have been removed as they were not sufficiently related to immunotherapeutic treatments.

Furthermore, per your third comment, the table was separated into two, one for mAb treatments and one for vaccines. Both have received titles and captions.

Lastly, the authors reviewed what was written concerning the challenges and limitations of immunotherapies and agreed that a more in-depth discussion of these concepts was warranted. Section 4.3 “The limitations and challenges of immunotherapies” was added, starting on line 274 and ending on line 295. This section covers challenges such as adverse events that can occur related to the immunotherapy administered, such as immune system overactivity that may exacerbate the condition or auto-immune response generation. Furthermore, the authors also discussed immunosenescence - age-related or otherwise - as a method by which immunotherapies may pose little-to-no effect, such as decreased production or activity of adaptive system lymphocytes.

Round 2

Reviewer 3 Report

Dear Authors,
Authors made necessary changes and it's getting close to the final acceptance.

However, authors must include the limitations of immunotherapies in AD as following:

1. Limitation of patients with CAA among AD patients, which was the main complication of side effects (ARIA) of Aducanumab and Lecanemab.

2. Immunotherapies in AD with abeta targeting antibodies were effective only at higher doses.

3. Crenezumab showed less ARIA complications, but was not effective.

4. Recommendations of use of immunotherapies should go along with use of biomarkers to support the efficacy or changes in patients.

Author Response

We appreciate your feedback and are glad to know that our changes were appreciated.

We have re-examined our review and have identified gaps in it that would benefit from the points you brought up regarding more specific limitations of immunotherapies, particularly related to ARIA complications and dose-dependency. As a result, we have added two new sections: 4.4. Analysis of immunotherapeutic efficacy (Lines 297-303) and 4.5. Cerebral amyloid angiopathy and Amyloid-related imaging abnormalities (Lines 305-323).

We hope these changes are viewed positively and look forward to any further feedback you may provide.